# Innovative Food Packaging, Food Quality and Safety, and Consumer Perspectives

**Mary R. Yan** [1,2,*], **Sally Hsieh** [3] **and Norberto Ricacho** [1]

1 School of Healthcare and Social Practice, Unitec Institute of Technology, Auckland 1025, New Zealand; nricacho@unitec.ac.nz
2 AUT Food Network, Auckland University of Technology, Auckland 1010, New Zealand
3 School of Environmental and Animal Sciences, Unitec Institute of Technology, Auckland 1025, New Zealand; shsieh@unitec.ac.nz
* Correspondence: myan@unitec.ac.nz; Tel.: +64-9-8928465

**Abstract:** Packaging is an integral part of the food industry associated with food quality and safety including food shelf life, and communications from the marketing perspective. Traditional food packaging provides the protection of food from damage and storage of food products until being consumed. Packaging also presents branding and nutritional information and promotes marketing. Over the past decades, plastic films were employed as a barrier to keep food stuffs safe from heat, moisture, microorganisms, dust, and dirt particles. Recent advancements have incorporated additional functionalities in barrier films to enhance the shelf life of food, such as active packaging and intelligent packaging. In addition, consumer perception has influences on packaging materials and designs. The current trend of consumers pursuing environmental-friendly packaging is increased. With the progress of applied technologies in the food sector, sustainable packaging has been emerging in response to consumer preferences and environmental obligations. This paper reviews the importance of food packaging in relation to food quality and safety; the development and applications of advanced smart, active, and intelligent packaging systems, and the properties of an oxygen barrier. The advantages and disadvantages of these packaging are discussed. Consumer perceptions regarding environmental-friendly packaging that could be applied in the food industry are also discussed.

**Keywords:** food packaging; food quality; food safety; applied technologies; consumer

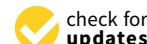

## 1. Introduction

Food processing involves a combination of various processes including preparation, (re)formulation, production, and packaging, etc. Packaging is an integral part of the food industry associated with food quality and safety including food shelf life, and communications from the marketing perspective. After food production and before reaching consumers, an important part is food packaging. The functions of traditional packaging are (i) holds food in packages for storage, transportation, and distribution, (ii) provides protection from physical damage and chemical and biological deterioration, (iii) stores food products until consumption [1–3], and (iv) also presents branding and nutritional information and promotes marketing [3,4].

In recent years, bioactive films and smart techniques have been applied in the food sector for packaging [5–7]. Plastic films, owing to their low weight, processing ease, and low cost have emerged as the most prominent candidate for food packaging applications [4]. Over the past decades, plastic films were employed as a barrier to keep food stuffs safe from heat, moisture, microorganisms, dust, and dirt particles. Recent advancements have incorporated additional functionalities in barrier films to enhance the shelf life of food, such as smart packaging [8,9], active packaging [10,11], and intelligent packaging [8,11–13].

These packaging systems provide a virtuous solution for extending the shelf life of food with simplified production processes or food with minimized use of preservatives [14,15].

In addition, consumer perception has influences on packaging materials and designs [11,16]. The trend of consumers demanding environmental-friendly packaging is increasing [17–19]. Moreover, nutrition and verifiable health-related information on the front- and back-pack could help shoppers to make healthier choices [20]. With the progress of applied technologies in the food sector, sustainable packaging has been emerging in response to consumer preferences and environmental obligations [21]. There have been a number of publications reviewing new techniques in packaging [1,5,11,22,23], however, consumer perception regarding renewable and environmental-friendly food packaging has not been clearly elucidated [24].

This paper reviews literature published mainly in the last 10 years that concerns the importance of food packaging in relation to food quality and safety; the applications of polymers (petroleum-based and bio-based) packaging; the development and applications of advanced smart, active, and intelligent packaging systems, and moreover, the properties of an oxygen barrier. Consumer perceptions regarding renewable and environmental-friendly packaging that could be applied in the food industry are also discussed in this review.

## 2. Polymers for Food Packaging

Polymers are commonly used materials for food packaging because of the easy production, strong molecular networks or crosslinking (polymetric matrices, Figure 1), and excellent performance characteristics (e.g., strength, the barrier to oxygen and moisture, and resistance to food component attack) [25–27]. Polymers are either non-biodegradable or biodegradable. For polymers that are commonly used in active food packaging, polypropylene (PP), polyethylene (PE), polyethylene-co-vinyl acetate (EVA), polyvinyl chloride) (PVC), and polyethylene terephthalate) (PET) are non-biodegradable polymers [28]. Cellulose, chitosan, starch, agar, gelatin, soy protein, and whey protein are biodegradable natural-based polymers [26,28] that have advantages over petroleum-based polymers from consumer and environmental perspectives.

The biodegradable, sustainable, and abundantly available polymers are categorized into polysaccharide type and protein-based type polymers [4,29,30]. Among polysaccharide-type polymers, cellulose is a polysaccharide consisting of a linear chain of β-1, 4 linked D-glucose units. Chitosan is a linear polysaccharide constituted by random β-1, 4 linked D-glucosamine, and N-acetyl-D-glucosamine. Cellulose and chitosan are widely used for food packaging due to their good film and gel-forming ability, recyclability, and inherent antimicrobial properties [28,31]. Starch is composed of linear and branched D-glucose units coupled by α-1, 4 and α-1, 6 glycosidic linkages; it can be used for food packaging as an adhesive and additive [28]. The commercially available starch and starch blends include Ecofram$^{TM}$, Solanyl$^{TM}$, Biocool$^{TM}$, Bioplast$^{TM}$, and Pantic$^{TM}$ [4]. Protein-type polymers, for example, soy protein, possess a wide range of functional properties to the protein-based films because of the intermolecular binding potential via covalent bonds [26]. Gelatin has strong film-forming abilities and is commonly used due to its abundance [4,32].

Polylactic acid blend (PLA) is a synthetic thermoplastic polyester. It is a commercially available environmental-friendly biodegradable polymer combined with antimicrobial agents to form PLA-based films used for the packaging of dry and perishable food products, usually short shelf-life products, such as fruits, vegetables, and meat [4,26,33].

## 3. Smart Food Packaging Systems

The fast-growing consumption of packaged food and beverages led to innovative packaging systems due to increased product complexity, food globalization, and consumer needs for environmental-friendly packaging [34,35]. As a result, smart packaging, active packaging, and intelligent packaging have emerged on the global market with their applications.

Smart packaging systems are widely used in food and beverages, healthcare products, personal care, and others, which can monitor physicochemical influences such as environmental conditions, and also prevent microbiological changes [12,23]. Active packaging and intelligent packaging that are discussed below in detail are types of smart packaging.

## 4. Active Food Packaging

An active packaging system is an advanced technology that embeds active components (e.g., antioxidants) into the polymeric packaging matrix (Figure 1) [36,37]. The polymeric matrix then releases or absorbs substances from or into the preserved food or the surrounding environment to sustain and prolong the shelf life of food. The system functions as a polymer matrix based on polymerization of polymers or biopolymers and bioactive agents releasing naturally to the food or the surrounding environment [15].

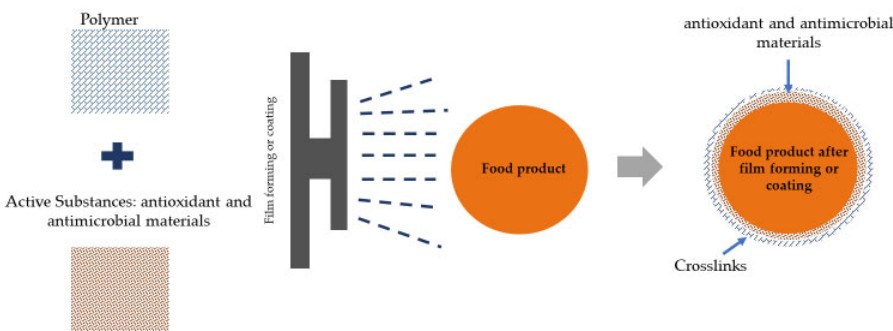

**Figure 1.** The system functions of a polymeric matrix, recreated from Munteanu and Vasile [37].

Active food packing presents an excellent potential in the maintenance of the quality and safety of food products, particularly oxidation-sensitive foods. Active agents for food packaging include antimicrobials, antioxidants, carbon dioxide emitters/absorbers, oxygen scavengers, and ethylene scavengers [23,38]. The commonly used agents and applications are presented in Table 1.

### 4.1. Antimicrobials

Antimicrobial packaging integrates antimicrobial agents into polymer film to suppress the targeted microbial activities [39]. Antimicrobial agents such as organic acids (e.g., benzoic acid), enzymes (e.g., lysozyme), bacteriocins, fungicides, and oxygen absorber (e.g., butylated hydroxytoluene) have been widely used in food packaging systems to prevent the growth of pathogenic microorganisms during the production, transportation, and storage of food products [39]. The advantage of the technique is controlled release as time-release or slow release of active substances over the postulated period. Antimicrobial packaging applies to preservative-free food products that are most likely to have pathogenic microorganism growth such as bread [40], cake, cheese, and meat [41,42].

Chemical additives are gradually replaced by natural compounds such as phytochemicals, bacteriocins, and enzymes, due to the increasing consumer demand for natural and safety assured food products. Natural antimicrobial agents contained in food stuffs have advanced applications in food packaging and are environmentally safer and effective, for example, essential oils extracted from clove, rosemary, oregano, lemongrass, basil, and fennel [22,43–45].

**Table 1.** Active agents for food packaging.

| Type | Commonly Used Agents | Applications |
|---|---|---|
| Antimicrobial | Chitosan<br>Essential oils<br>Gallic acid<br>Lactoferrin<br>Lysozyme<br>Metals<br>Nisin | Fruits [46], bread [40], meat [41,42,44,47,48], fish [49,50] |
| Antioxidant | Essential oils<br>Lignin<br>Plant extracts<br>Phenolic compounds<br>α-Tocopherol | Cereals, nuts, meat, meat products [41,44,51] |
| Carbon dioxide absorber/emitter | Citric acid<br>Ferrous carbonate<br>Sodium bicarbonate | Meat, fruits, vegetables [52] |
| Oxygen scavenger/absorber | Ascorbic acid<br>Gallic acid<br>Glucose oxidase<br>Iron<br>Laccase<br>Palladium<br>Pyrogallol | Most baked products and nuts [53], meat [38,53], fish [53], fruits [54] |
| Ethylene scavenger | Activated carbon<br>Potassium permanganate<br>Metal oxides<br>Metal organic frameworks (MOFs)<br>Titanium dioxide | Fruits, e.g., kiwifruit, banana, vegetables [22] |

### 4.2. Antioxidants

Antioxidant agents inhibit unwanted microbiological changes and oxidative reactions to extend the shelf life of food. Using natural antioxidants in food packaging is a new trend in the food industry in response to increasing consumer demand for natural products [55].

Plant extracts from plant stems, roots, leaves, and fruit seeds have been effectively used as antioxidant components in food packaging [30,56]. Plant extracts contain considerable content of polyphenols, flavonoids, alkaloids, and terpene substances that have proven antioxidant properties. Plant extracts added to packing materials are applied in the form of films.

Essential oils from plants such as cumin, fennel, mint, thyme, rosemary, cinnamon, onion, and garlic have been recognized as antioxidant agents [30,43–45]. These compounds are incorporated into the biopolymer matrix, for example, gelatin and gelatin-montmorillonite films to improve the antioxidative properties [30].

Phenolic compounds such as polyphenol, flavonoids, and quinones from plant extracts can act as both antimicrobial and antioxidant agents [57].

Plant-derived antioxidants incorporated with active packaging inhibit food spoilage and deterioration without direct contact with foods and changes in sensory quality [58].

### 4.3. Oxygen Scavengers

Oxygen residual in food packages causes bacterial spoilage, off-flavor development, color change, nutrient loss, and toxic end-products forming [22,53]. Oxygen scavengers or oxygen absorbers are added to enclosed packaging to remove or reduce the level of oxygen in the packaging to inhibit the formation of aerobic pathogenic microorganisms and oxidation, and maintain the quality of high unsaturated fat-containing food [53]. Oxygen

scavengers or oxygen absorbers are commonly used for packaged food products and pharmaceutical products [53,59] (Figure 2).

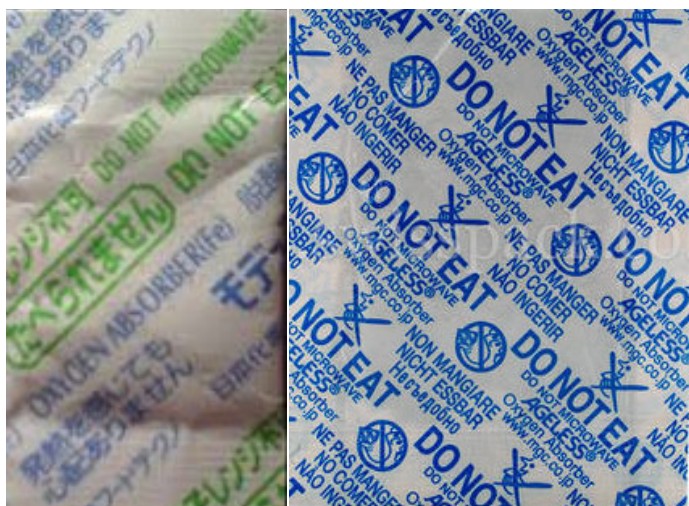

**Figure 2.** Examples of oxygen absorbers.

### 4.4. Carbon Dioxide Scavengers

The role of carbon dioxide ($CO_2$) used in food packaging is in relation to the protection of foods from oxidation and its antimicrobial effects [52]. Carbon dioxide has the potential to inhibit microbial growth and is highly soluble in food matrices in modified atmosphere packaging. Modified atmosphere packaging (MAP) is a packaging technique that involves either actively or passively controlling or modifying the gas composition of the food storage environment to reduce food oxidation and the growth of aerobic spoilage organisms [60]. Passive MAP is achieved when the desired atmosphere is developed naturally through a food product's respiration. Active MAP is achieved by replacing gases in the packaging with a desired mixture of gases [61]. The carbon dioxide scavenger technique is used in combination with food refrigeration for fresh and fermented food products. The commercial availability of active packaging is still limited due to the low achievement on an industrial scale and the low stability of the active agents [51].

### 4.5. Ethylene Scavengers

Ethylene scavengers are used for storing fruits and vegetables [22], as ethylene leads to undesirable color changes and off-flavor which shortens the shelf life of fruits and vegetables during postharvest and processes.

### 5. Intelligent Packaging

An intelligent packaging system indicates and monitors the physicochemical conditions of the product (e.g., the degree of freshness), and its environmental influences (e.g., temperature, pH level, gas) during transportation and storage [13,62,63]. The device designed for the packaging can sense any changes internally or externally, and can then inform the consumer regarding the status of the product by providing sound and visual information (Figure 3) [22,62,64]. The central part of intelligent packaging is integrating sensors or indicators, or radio frequency identification systems (RFID) (Table 2).

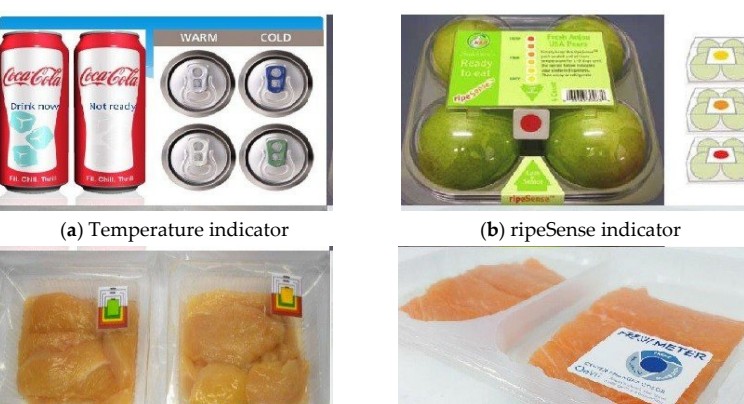

(**a**) Temperature indicator          (**b**) ripeSense indicator

(**c**) Freshness indicator          (**d**) Freshness indicator

**Figure 3.** Applications of intelligent packaging systems, modified from Nešić et al. [64]. (**a**) Temperature indicator; (**b**) ripeSense indicator; (**c**) Freshness indicator [65]; (**d**) Freshness indicator.

**Table 2.** Different types of intelligent packaging.

| Type | Systems | Applications |
|---|---|---|
| Sensor | Bio-sensor<br>Gas sensor<br>Fluorescence-based oxygen sensor | Meat [66], fish, fruits [62,67], beverages, vegetables [68,69] |
| Indicator | Time/temperature indicator | Food stored under chilled and frozen conditions (e.g., meat [38], milk [70]), pharmaceutical products [71] |
| | Oxygen indicator | Food stored in MAP (e.g., meat [38]) |
| | Carbon dioxide indicator | Food stored in MAP or CAP [72,73] |
| | Color indicator | Fish [74], milk [75] |
| | Pathogen indicator | Meat [73], fish [76,77] |
| | Breakage indicator | Canned baby foods [8] |
| | Leak indicator | Perishable foods [67] |
| | Freshness indicator | Fish [76,77], poultry [65], fruits [67] |
| Radio frequency identification tag (RFID) | | Cheese [78], meat [22], vegetables [69] |

MAP: Modified atmosphere packaging; CAP: Controlled atmosphere packaging.

*5.1. Sensors*

An intelligent sensor (e.g., bio-sensor, gas sensor) is a device that comprises a receptor transforming chemical or physical information into a form of energy, and a transducer transforming the energy into a useful analytical signal. The device can locate, detect or quantify matter or energy, then send signals for the detection of a chemical or physical property to which the device responds [5,8]. Bio-sensors detect, transmit, and record information about biological reactions that include bioreceptor (specific to a target analyte such as microbes, hormones, enzymes, antigens) and transducer (to convert biological signals to an electrical response such as electrochemical, optical) [62]. Gas sensors respond to the presence of a gaseous analyte in the package quantitatively and reversibly, and then change the physical parameters of the sensor, and are monitored by an external device [23]. There are commercially available gas indicators: Ageless Eye$^{TM}$, Shelf Life Guard, Tell-Tab, and Tufflex GS [5,63].

*5.2. Indicators*

Intelligent indicators are substances that give quality-related information on temperature, leakage, generation and concentration of carbon dioxide, microbiological status, freshness, and appearance or color [23,62].

Moreover, time-temperature indicators give information about temperature and show the variation and history in temperature change [38,70]. Oxygen indicators give information on leakage and quality deterioration of MAP foods [38,79]. Color indicators give information through tags attached as small adhesive labels to the outside of food packaging [80]. Pathogen indicators give information on microbiological status [38]. Breakage indicators give information on broken packaging. Freshness indicators detect the growth of pathogenic microorganisms, and provide information on the quality of the products related to spoilage and microbial growth [38,76,81]. Fresh Tag®, Sensor Q™, Food Sentinel System, and Toxin Guard® are commercially available freshness indicators [5,63].

### 5.3. Radio Frequency Identification (RFID)

Radio frequency identification (RFID) is wireless communication based on tags and readers placed on containers and pallets to gather real-time information about temperature, relative humidity, shelf life, and nutritional information through the supply chain management [11,13,63]. The commercially available RFID include Easy2log®, Intelligent Box, and Temptrip [23].

The intelligent packaging system has the advantage to provide information, extend shelf life, improve the quality of food, and enhance safety. However, it adds additional cost onto the final cost of food products, in particular, RFID system [13], further with a risk of potential migration of the packaging particles into food products [11].

## 6. Oxygen Barrier Properties

Oxidative deterioration is one of the major concerns associated with the shelf life of food products. Adding antioxidants is an economical and effective physiochemical solution. The oxygen barrier techniques are the applications of polymers in food packaging [82,83]. The function of the polymer barrier is to stop oxygen permeation though the food packaging (Figure 4). The commercially used oxygen barriers include ethylene-vinyl alcohol copolymer (EVOH), polyacrylonitrile (PAN), polyketone (PK), and polyethylene terephthalate (PET) [82]. Oxygen barrier guards, for example, a PET-based oxygen BarrierGuard, are applied for oxidation-sensitive food products including nuts, oils, baby foods, coffee, and vegetables [84].

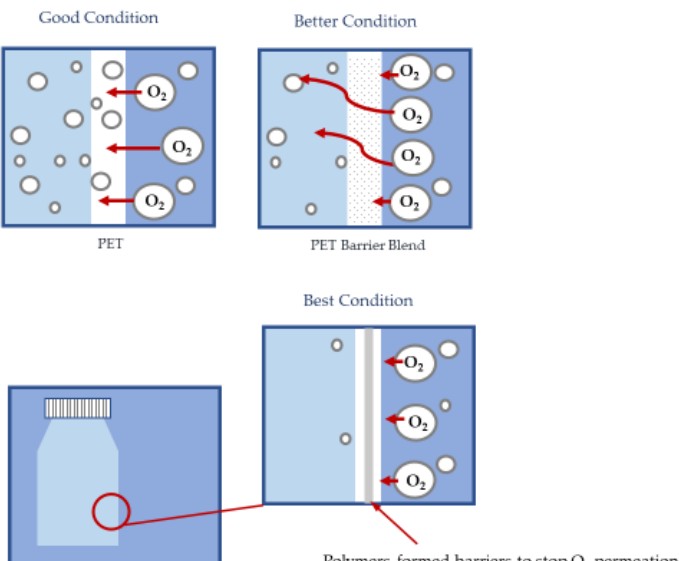

**Figure 4.** Efficient oxygen barrier guard, recreated from Lingle [84].

Polysaccharide-type polymers formed films and coating which are good oxygen barriers associated with the large number of hydrogen bonds that assist adjacent chains to bind tightly to each other [85]. Protein-based polymers are also good oxygen barriers and can carry active substances that could migrate to food surfaces [85,86].

## 7. Consumer Perceptions and Edible Films

The origin, production type, and packaging are the three most important indicators that evaluate the environmental impact of food from a consumer survey ($n$ = 797) [35]. With the increase of awareness of eco-sustainable consumption, consumer demands for environmental-friendly packaging are becoming a new trend in the food sector [17,19,87]. The packaging materials and information displayed on packaging influence consumer expectations and willingness to purchase [20,88]. Consumers prefer food packaging using additives from natural sources and innovative materials that can be recycled and beyond the primary fundamental role of the protection and preservation of food products [89]. Edible films and coatings, e.g., starch-based and protein-based, are good examples [85,90–92].

The initiative of edible films and coatings is related to the natural antimicrobial agents and bioactive polymers contained in carbohydrates or protein in foods (Table 3) [45]. For instance, commonly used materials for edible films and coating include nisin [45,93,94], soy protein and gelatin [32,95], chitosan [31,57], starch [96,97], corn starch [98], rice starch [99], agar [92], and carrageenan [100]. They are environmental-friendly green polymers [19]. Agar-based edible films containing functional agents (e.g., green tea extract, cinnamon essential oil) inhibit the growth of food-borne pathogens, improve the quality and extend the shelf life of food. Starch-based films with the addition of essential oils is a concept of sustainable food packaging from renewable sources [101]. Chitosan films are used as edible films and coating [45]. Soy protein films coated with nisin and natural plant extracts (e.g., grape seed extract, green tea extract) can be used for ready-to-eat meat packaging [45].

**Table 3.** Natural materials for food packaging.

| Form | Substances | Applications |
|---|---|---|
| Active natural-based edible film | Agar | Fruits, vegetables [4], chicken, fish [92] |
| | Agar-nisin | Fish [102] |
| | Chitosan/cellulose | Chicken [103], ham [48,104], fruits [4,105] |
| | Starch/corn starch | Fruits [106], beef [96,98], meat [107] |
| | Soy protein and gelatin | Meat [45], fish [80], fish oil [92] |
| | Whey protein | Lamb [108] |
| | Propolis (bee glue/resin) | Fruits, vegetables, meat, fish [109] |

## 8. Discussion and Future Prospects

The most recent advances in food packaging have been the development of active packaging and intelligent packaging, and the development of biodegradable polymers, edible films and coatings [110,111]. From economic considerations, innovative food packaging reduces food waste or loss of food quality by improving food shelf life. Additionally, utilizing of renewable biodegradable packaging materials minimizes environmental impacts. It also benefits consumer health [11,21].

An active packaging system incorporates active components into packaging to improve the quality and shelf life of foods, and an intelligent packaging system monitors the conditions and provides information related to the quality of packaged foods [10].

Active packaging has the following advantages: prevents oxidation, microbial growth, and color loss, removes off-flavor, and slows down metabolism of food. Disadvantages include increasing the cost of production, possible migration of complex packaging materials into food, lack of recyclability of disposable packaging, and cannot be allied in liquid food.

The advantage of intelligent packaging systems is that they can be integrated into packaging and checked by the naked eye [66], can monitor the external conditions continuously [22], provide information about the condition of food, warn about possible problems,

reduce food loss, and detect calamities during transportation. The disadvantages include high cost and lack of recyclability [13].

Natural antimicrobial and antioxidant agents (e.g., plant extracts, essential oils) are a favorite consumer preference and good for the ecosystem, however, in vitro studies are needed to evaluate the safety and possible side effects when used in food [112]. In addition, compared to plastic packaging materials, bio-based films are biodegradable, but have low elasticity, low thermal stability, and high water sensitivity [92], thus, further study is needed in improving their functional properties.

For all these packaging systems discussed, they need to be inexpensive relative to the value of the product, reproducible, consumer acceptable, have no negative effects on health, and be environmental-friendly [92]. Various factors could influence the implementation of active and intelligent packaging, such as market-driven factors, the gap between science and industry [30], and the gap between industry and consumers [113]. The bioactive, biodegradable, and bio-nanocomposite (3-bios) combined strategy would be an innovative concept for the future trend of food science and nutrition [87].

## 9. Conclusions

In conclusion, innovative food packaging provides a virtuous solution for the food industry in the maintenance of food quality and safety. Further research would fill the gaps as mentioned above to fulfil large-scale commercial reality, minimize the impact on the environment, and meet consumer acceptance.

**Author Contributions:** M.R.Y. wrote the manuscript and approved the final draft to be published. S.H. and N.R. provided constructive feedback. All authors have read and agreed to the published version of the manuscript.

**Funding:** This research received no external funding.

**Acknowledgments:** The APC is funded by Tuapapa Rangahau, Unitec Institute of Technology.

**Conflicts of Interest:** There is no conflict of interest.

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
