# Peer review of "Innovative Food Packaging, Food Quality and Safety, and Consumer Perspectives"

_processes, doi:10.3390/pr10040747_

Round 1
Reviewer 1 Report
The review work is solid and presents the overview it proposes. The authors may add an analysis of the number of publications in the last 20 years, in order to justify the importance of this review.
I also suggest that the authors add a section discussing the most recent advances in the field, as well as a section analyzing the economics aspects of packaging.
Author Response
The review work is solid and presents the overview it proposes.
Thank you very much for your support and comments, that helps to improve the quality of our manuscript.
The authors may add an analysis of the number of publications in the last 20 years, in order to justify the importance of this review.
Line 58, the sentence has been changed to ‘This paper reviews literature published mainly in the last 10 years that concerns the importance …’.
I also suggest that the authors add a section discussing the most recent advances in the field, as well as a section analysing the economics aspects of packaging.
Thank you very much for your suggestion. The most recent advances in food packaging and the economic considerations have been included in Discussion (see page 9, lines 274-279).
Reviewer 2 Report
A well prepared article however one part is missing in packaging sections: Modified air packaging. examples; https://doi.org/10.1108/BFJ-06-2018-0408
Author Response
Thank you very much for your support and suggestion that helps to improve the quality of our manuscript. Modified air packaging has been added to the context (see page 5, lines 170-175).
Reviewer 3 Report
Innovative food packaging, food quality and safety, and consumer perspectives
Comments
- Overall, try to write a short sentence which should not be more than three lines otherwise, its look like a paragraph.
- L-18 modify this line.
- L-50 replace word wanting with demanding.
- L-99 missing reference.
- L-105 replace word system.
- L-112 missing reference.
- L-196 CAP Controlled packaging replaced with controlled atmosphere packaging.
- L-201 modify this sentence.
- L-203,204 Color indicators give information on temperature in food packaging.
It should be replaced with (Color indicators gives information through tags attached as small adhesive labels to outside of food packaging).
- L-261 conclusions and further prospects should be without citations.
- L-274 add preposition in this sentence.
- Improve bibliography according to required journal guidelines.

Author Response
Thank you very much for your support and comments, that helps to improve the quality of our manuscript.
- Overall, try to write a short sentence which should not be more than three lines otherwise, its look like a paragraph.
Thank you for your comments.
Lines 34-37, the sentence has been rewritten.
Lines 69-74, the sentence has been divided into two.
- L-18 modify this line.
Line 18, ‘for examples’ has been deleted.
- L-50 replace word wanting with demanding.
This has been done.
- L-99 missing reference.
References [12,23] about smart packaging have been added (see line 101).
- L-105 replace word system.
‘system’ has been replaced with ‘polymeric matrix’ (see line 106).
- L-113 missing reference.
References [23,38] about active packaging have been added (see line 116).
- L-196 CAP Controlled packaging replaced with controlled atmosphere packaging.
Thank you for your comments, this has been done (see line 205).
- L-201 modify this sentence.
Thank you for your suggestion. This has been done (see lines 210-211).
- L-203,204 Color indicators give information on temperature in food packaging.
- It should be replaced with (Color indicators gives information through tags attached as small adhesive labels to outside of food packaging).
Thank you very much for your suggestion, that has been changed (see line 213).
- L-261 conclusions and further prospects should be without citations.
Thank you very much for your comments, we have divided the last section into two: Discussion and future prospects, Conclusions (see page 9).
- L-274 add preposition in this sentence.
This has been done (see line 292).
- Improve bibliography according to required journal guidelines.
Thank you for your comments, this has been done.